# Photosystem II Performance of *Coffea canephora* Seedlings after Sunscreen Application

**DOI:** 10.3390/plants12071467

**Published:** 2023-03-27

**Authors:** Débora Moro Soela, Edney Leandro da Vitória, Antelmo Ralph Falqueto, Luis Felipe Oliveira Ribeiro, Cátia Aparecida Simon, Luciano Rastoldo Sigismondi, Rodrigo Fraga Jegeski, Leandro Demetriu Becatiini Pereira

**Affiliations:** 1Postgraduate Program in Tropical Agriculture (PPGAT), Federal University of Espirito Santo (UFES), São Mateus 29932-510, ES, Brazil; 2Department of Agricultural and Biological Sciences (DCAB), Federal University of Espirito Santo (UFES), São Mateus 29932-510, ES, Brazil; 3Department of Agronomy, Integrated Faculties Espírito-Santenses (FAESA), Linhares 29900-070, ES, Brazil; 4Department of Agronomy, Faculty Pitagoras, University Pitagoras, Linhares 29900-070, ES, Brazil

**Keywords:** application technology, photoprotection, Kautsky curve, sunscreen for plants, abiotic stress, photosynthetic activity, calcium carbonate-CC, conilon

## Abstract

In the conilon coffee tree, the stress caused by high light can reduce the photosynthetic rate, limit the development and also reduce the yield of beans. Considering that the quality of a sunscreen spray can influence photosynthetic performance, the goal was to understand the iterations between the quality of the spray and the variation of the chlorophyll *a* fluorescence when applying sunscreen on conilon coffee trees. The parameters coverage, volumetric median diameter, density, droplet deposition, and the variation of the chlorophyll *a* fluorescence were evaluated. The nozzle and application rate factors did not show direct effects in the physiological responses of the plants. Plants with no sunscreen application showed high values of energy dissipation flux. The photosystem II (PSII) performance index and PSII photochemical maximum efficiency indicate that the use of sunscreen for plants promotes better performance of photosynthetic activity and that it provides photoprotection against luminous stress, regardless of the application rate and spraying nozzle; however, we recommend using the application rate of 100 L ha^−1^ and the cone jet nozzle type because they provide lower risks of product loss due to runoff.

## 1. Introduction

Coffee is the second most traded food *commodity* in the world, only behind crude oil [1]. Considering the 2021–2022 harvest, world production was approximately 167.5 million 60 kg bags, with an 11% increase forecast for the 2022–2023 harvest [2]. In Brazil, the production of conilon coffee (*Coffea canefora*) for the 2021–2022 crop had an increase of 11.7% when compared to the previous crop, equivalent to a total of 18,199.3 thousand processed bags, 12,358 thousand bags of which were produced in the state of Espirito Santo, which is the largest producer of conilon coffee in the country, representing 68% of the national production [3].

Asexual propagation in conilon coffee is responsible for approximately 90% of its seedlings produced because they present characteristics similar to the mother plants, causing the development of more homogeneous plantations, with desirable features in terms of canopy architecture, establishment of more vigorous productive branches, plants with good productivity, uniform harvesting and resistance to diseases [4,5].

In the nursery, the production of seedlings is normally done in a shaded environment, undergoing a gradual acclimatization process before transplanting to the field, facilitating their adaptation and establishment. The acclimation takes place after the second or third pair of leaves emerge and is the result of the gradual removal of the shade cloth from the nursery area until the plants are completely exposed to full sunlight [6,7]. Thus, it is expected that the photosynthetic apparatus will acclimatize to the high light condition, which is an important factor determining the plant fitness or crop yield [8].

The climate changes that took place over the last decades and the need for adaptation of the coffee culture may interfere in the development of seedlings and, consequently, negatively influence future productivity. High light directly affects the plant morphophysiology, including changes in leaf and vascular diameter and thickness, and also in the physiological and biochemical processes [9]. Adjustments to the composition of reaction centers (RC) and light harvesting antennae are related to enhanced chloroplast physiology. As a result, the ability of seedlings to acclimatize to varying light condition may affect plant growth efficiency [10].

However, excess light can cause damage to the photosynthetic apparatus, especially PSII. The accumulation of photodamaged PSII decreases photosynthetic activity impairing the physiological capacity of seedlings [11], which reduces the electron transport rate through of PSII and, consequently, the photosynthetic efficiency, which can be used as an indicator of the photoinhibitory damage in the PSII complex, altering the transient OJIP shape, caused either by decreases in Fm or by increases in F0. This phenomenon is referred to as photoinhibition, characterized by high dissipation of energy as heat (DI0/RC) associated with low conservation of energy in the PSII (PIABS), which could potentially be converted into chemical energy (ATP and NADPH) [12].

In order to avoid damage to the photosynthetic apparatus, the application of some chemical compounds may result in protection against heat and light stress and, consequently, improve plant performance in the field. Calcium carbonate (CC) compounds have been used as an efficient technology to increase plant adaptation to adverse climate by controlling photoxidative stress [13]. The use of CC was associated with photoprotective effects on grapevines [14], leaf temperature reduction in apple trees [15] and photoprotection followed by increased net photosynthesis rate in coffee [13]. After transplanting the seedlings, high levels of irradiance caused photo-oxidation and scalding symptoms on the leaves of the coffee plants. The previous application of sunscreen allows a well-formed seedling to carry with it all the nutrient reserves and a structure with the strength to withstand adversity when it is planted in the field [13,14,15].

However, there is an evident gap in the application processes of products considered sunscreens because there is no specific work related to pesticide application technology regarding application efficiency and its relation to the effectiveness of the product applied. The process of applying agricultural products (pesticides, foliar fertilizers and similar) through the spraying of the mixture applied is extremely important because an efficient application can optimize the deposition on the target, ensure the effectiveness of the product, no waste through runoff and minimizing the effect of drift [16,17].

In the specific case of the application of a product that functions as a sunscreen, the efficiency of spraying is related to the ability of the droplets to be deposited on the leaves forming a protective film in order to enhance the photochemical effect of sunscreen. In this sense, the final amount of product that adheres to the leaf after spraying is defined as a function of coverage, density and size of the drops deposited. These variables depend on the type of spray nozzle, the working pressure and travel speed [18,19].

The volume of sunscreen applied (volume per area) depends on the crop development stage, the location of the target in the crop and the equipment to be used for spraying and sunscreen application [20,21,22]; in addition, the choice of spray nozzle and working pressure influence the kinetic dynamics of the deposition on the target.

Therefore, considering the hypothesis that the photochemical effect of sunscreen applied on conilon coffee seedlings depends on the application technology, this work aims to understand the iterations between the spray quality and the variation of the chlorophyll *a* fluorescence with the application of a sunscreen based on CC and zinc oxide on conilon coffee seedlings.

## 2. Results

The average droplet densities deposited in the 1st and 2nd sunscreen applications are presented in Table 1. The interaction between the nozzles and the application rates were significant in both moments of sunscreen application, indicating the dependence between the two factors.

The flat spray nozzle BD 015 provided higher density of drops compared to the empty cone spray nozzle, a little more than twice in both moments of application, regardless of the application rate. The empty cone spray nozzle produces smaller droplets compared to the flat spray nozzle, which can provide more effective droplet distribution; however, there is potential drift risk.

The values of droplet density deposited when using the MGA 60 nozzle increased 18.9% and 19.9% when the application rate was increased by 50 L ha^−1^ in the 1st and 2nd applications, respectively. The increase in droplet density was also observed by varying the application rate from 100 to 150 L ha^−1^, with the use of the BD 015 nozzle, being 21.0 and 28.8% in the 1st and 2nd applications, respectively. It is worth mentioning that higher values of drops deposited indicate higher values of sunscreen retained by the leaves; however, from the moment that the leaf no longer retains the sprayed sunscreen, there may be runoff and consequent loss of the product applied to the soil.

The behavior of the coverage was similar to the droplet density; that is, a significant iteration was observed between the spray nozzles and application rate factors, therefore indicating the dependence of these factors in the application of the product at the two application times (Table 2).

Increasing the provided application rate increased coverage, regardless of the spray nozzle and at both application times. The coverage was 1.17 and 1.14 times greater for the MGA 60 01 nozzle for the 1st and 2nd applications, respectively, and 1.06 and 1.14 times greater for the BD 015 nozzle for the 1st and 2nd applications, respectively. Although the spray droplet sizes were different between the nozzles, which according to the manufacturer are very thin for the MGA nozzle and thin for the BD nozzle, this difference did not translate into different coverage values. The risk of drift could influence the coverage on target as well as the deposition.

When comparing the nozzles at the same application rate, there is a significant difference in the average coverage values, with the BD 015 nozzle showing higher coverage values at both application times. Although the risk of drift was not one of the parameters analyzed, the step variation in wind intensity might have contributed to the lower coverage observed for the MGA 60 01 nozzle.

The effect of nozzles and application rates on volumetric median diameter during sunscreen application is shown in Table 3. The interaction between these factors was not significant, indicating the independence of the factors. Regardless of spray nozzle and application timing, no significant difference was observed for the application rate factor. Normally, as you increase the pressure and therefore the application rate, the droplet size is reduced. However, the small difference in application rate applied in the experiment was not enough to attest to this trend.

The BD 015 spray nozzle provided significantly larger median diameters compared to the MGA 60 01 nozzle. In the first application, the volumetric median diameter was 18.7 and 16.5% larger with the BD nozzle compared to the MGA nozzle at application rates of 100 and 150 L ha^−1^, respectively. In the second application, at application rates of 100 and 150 L ha^−1^, the volumetric median diameter values were 13.3 and 12.2% larger with the BD nozzle compared to the MGA 60 01 nozzle. The results were expected because the MGA 60 01 spray nozzle produces very thin droplets and the BD nozzle produces thin droplets at the same test pressure indicated by the manufacturer. The average deposition results are shown in Table 4.

The iteration between the application rate and nozzle type factors was not significant; the significant difference was observed only for the nozzle variable. The flat spray nozzle BD 015 provided higher average deposition values for the same application rate, both in the first and second application. In the first application, the average differences in deposition between the compared nozzles were 1.34 and 1.44 µg cm^−2^, at the rates of 100 and 150 L ha^−1^. In the second application, the average difference between the nozzles was 1.04 µg cm^−2^ for the 100 L ha^−1^ rate and 1.16 µg cm^−2^ for the 150 L ha^−1^ rate.

The OJIP curves obtained after spraying conilon coffee seedlings with sunscreen when the different spray nozzles were used showed a typical polyphasic behavior with well-defined O, J, I and P-steps, indicating that the plants remained photosynthetically active (Figure 1A,B). On the 7th day there was greater homogeneity in the OJIP transient steps (Figure 1A), meanwhile regarding the second evaluation performed on the 30th day of exposure to light stress. The results indicate that, in the control treatment, there was suppression of phases J-I and I-P of the OJIP curve (Figure 1B), evidencing the occurrence of partial or total blockage of the energy flow in the electron transport chain [23].

The JIP-test parameters can be used to characterize the behavior of PSII in response to different types of external environmental stresses, such as heat and light intensity [24]. In this study, the variables describe about the absorbed energy flux (ABS/RC) and the energy flux captured by CR (reaction center) at t = 0 (TR_0_/RC) did not differ (*p* > 0.05) between control and treatments after 7 days of sunscreen application in the different spray nozzles. However, increased values (*p* ≤ 0.05) of ABS/RC and TR_0_/RC were obtained after 30 days of sunscreen application on conilon coffee seedlings (Figure 2A,B).

In this study, RC/CS_0_ values were similar (*p* > 0.05) after 7 days of treatment (Figure 3). However, at 30 days, the sunscreen-treated plants presented higher RC/CS_0_ values compared to control plants (*p* ≤ 0.05) (Figure 3).

The values for electron transport flux per RC at zero time (ET_0_/RC) were different between treatment in both 7 and 30 days of exposure to high light (Figure 4A). However, after 7 days, higher ET_0_/RC values were observed in the control plants (without the applications of CC). In contrast, after 30 days, the use of CC caused an increase of ET_0_/RC values compared to control plants (Figure 4A). Also, the use of CC reduced the values of dissipated energy flux per RC at t = 0 (DI_0_/RC) ratio in coffee plants after 30 days of exposure to high light after application of CC (Figure 4B). No change was observed after 7 days (*p* > 0.05).

The performance index of PSII (PI_ABS_) values were different between treatment and control at 7 and 30 days after application (Figure 5A). After 7 days, higher PI_ABS_ was observed for the control plants compared to those receiving CC. After 30 days of experiment, the use of calcium carbonate increased PI_ABS_ (Figure 6A). Finally, the maximum quantum efficiency of PSII (Fv/Fm ratio) increased significantly (*p* ≤ 0.05) with the application of CC after 30 days (Figure 5B). No change of Fv/Fm ratio was observed after 7 days (*p* > 0.05).

## 3. Discussion

The presented results show that the sunscreen application rate associated with the type of spray nozzle on conilon coffee seedlings has a significant effect on droplet distribution. The average values of coverage and density increase significantly due to the increased application rate. The increase in the application rate, that is, the increase in the volume applied per unit area considered for the crop in question, directly provides greater coverage and density of drops because this parameter is directly related to the increase in volume sprayed that, associated with the type of nozzle and working pressure used, has an influence on the quality of the application. It was observed that the flat spray nozzle provided higher average values in the four variables analyzed (density, coverage, volumetric median diameter and deposition); the fact that this type of nozzle produces medium droplets explains the results. The average values increase for the four response variables when using the flat spray nozzle over the empty cone type nozzle does not determine better spray quality because the exceeded amount of product applied by the first nozzle has high runoff potential, causing part of the sunscreen applied in the spray not to remain on the leaves. Similar results are reported [21,25], and these authors also verified the potential for drift and runoff of the spray sunscreen applied with increasing application rate.

The volumetric median diameter and droplet deposition increases significantly when comparing the flat spray nozzle to the empty cone nozzle, regardless of the application rate used. The flat spray nozzles have a larger orifice that increases the kinetic energy of the droplets sprayed, which in case of seedlings with little foliage can be harmful, because the higher kinetic energy can reflect a high impact of the droplet with the target and the droplets with the product can be lost by runoff. Cone jet nozzles, on the other hand, project fine droplets with lower kinetic energy that have a greater ability to be retained on the target; however, there is the disadvantage that these fine droplets can be lost through drift or evaporation depending on climatic conditions. Larger droplets decrease the risk of product drift but have less coverage on the leaf. This explains why the cone type nozzles showed a better result in droplet density and coverage. According to [16,26], the large drops have a greater weight, and due to this factor, it is difficult to set on leaf surfaces, lacking uniformity and coverage of drops.

Regarding the photochemical responses, the relation between application rates and nozzle type had no significant influence, what allowed the analysis of these variables to be performed with the average values of the treatments compared to the control treatment. However, the average variable values of droplet density, coverage, volumetric median diameter and deposition indicate that spray quality is influenced, as excess droplets deposited at a higher application rate and by the flat spray nozzle can cause runoff and product waste on the target. Since there is no homogeneity of the droplets, the risk of losses increases [20].

The chlorophyll *a* fluorescence measurements can show us damage to the photosynthetic apparatus, through changes in the amplitude of the OJIP curve, as described by [26]. According to [27], the initial fluorescence measurement (F_0_) is indicative of the time when all RCs are oxidized while the maximum value, or P-step (F_M_), occurs when all RCs are reduced (closed).

Phase I-P indicates the estimated rate of reduction of the pool of PSI-associated electron end acceptors [28,29]. The results obtained in this study showed an expressive increase in the magnitude of fluorescence in the I-P phase at 30 days after the sunscreen application. This phase showed the greatest influence on the photosynthetic activity of the plants, allowing us to assume that the application of sunscreen promoted an increase in the size of the pool of electron acceptors in the PSI.

On the other side, the suppression of the I-P phase observed for plants that did not receive CC application indicates a reduction of the final electron receptors on the acceptor side of the ISF, i.e., ferredoxin, other intermediates and NADP+, consequently reducing the synthesis of reducing power for CO_2_ assimilation [30]. Furthermore, the reduction in fluorescence intensity at the P-step (F_M_) observed when calcium carbonate was not applied is related to the accumulation of unreduced Quinone A (Q_A_) by the electron donor side of PSII, functioning as a suppressor of fluorescence [31]. Other authors also observed reduced P-step in plants of *Plectranthus scutellarioides* and sugarcane subjected to water stress [32,33].

These results, when associated with the lower ETo/RC and PI_ABS_ values as well as the higher DIo/RC values recorded in the absence of sunscreen application, show a reduction in the electron transfer efficiency from Quinone A to Quinone B (Q_A_–Q_B_). According to Campostrini [34], the increase in fluorescence signals at I-step reveals a decline in the photochemical process, caused by the reduction of the Q_A_ acceptor. Phase J-I is characterized by the onset of electron transfer from Q_A_ to the plastoquinone pool via Q_B_. Thus, an increase in the J-I phase occurs due to the increase in the concentration of reduced Q_A_ and Q_B_. Then, after the P-step (phase I-P), Q_A_ is partially oxidized, and electron transfer to PSI via cytochrome b6/f and plastocyanin complex occurs. The increase in the P-step is related to the maximum concentrations of Q_A_^−^, Q_B_^−^ and the electron transport from Q_B_^−^ to plastoquinone-PQ [35,36]. Thus, the application of CC improved the passage of electrons from reduced plastoquinone (PQH_2_) to the final electron receptors of the PSI in an efficient manner, resulting from the functioning of the photosynthetic apparatus [37].

The specific light absorption flux per reaction center (ABS/RC) is a measure of the effective size of the antenna system, defined by the ratio of active and inactive CRs, where the number of photons absorbed by the chlorophyll molecule is divided by the active CRs [23,38]. Thus, the increase in the ABS/RC ratio obtained for control plants may be related to some hypothesis, such as: (1) growth in the size of the antenna which provides excitation energy to the active RCs and (2) part of the RCs are inactive [29,30]. The results obtained for the variable RC/CSo confirm the hypothesis that the lower ABS/RC values observed after 30 days of CC application occurred due to the increase in active RCs (see Figure 3B) [26].

On the other hand, the inactivation of RCs may indicate greater susceptibility to photoinhibition and, consequently, reduced CO_2_ assimilation rate, as well as a direct relation with the increase in energy dissipation flux per reaction center (DI_0_/RC), because, when RCs are inactivated, the energy trapping and transport are impaired (TR_0_/RC and ET_0_/RC, respectively) [39]. Thus, this energy is dissipated in the form of heat as observed by higher DI_0_/RC reported to coffee plants that did not receive the application of CC after 30 days [40,41]. However, the reduction of DI_0_/RC obtained after the application of sunscreen shows that the product has photoprotective action on the plant, i.e., promotes mechanisms of better efficiency of absorbed energy utilization and functioning of the electron transport chain. According to [13], the application of CC to coffee trees was an effective alternative to promote artificial shading and to alleviate light and heat stress on the plants.

The ratio Fv/Fm or PSII maximum photochemical efficiency is a sensitive parameter to identify plant stress [42,43,44,45]. The appropriate values for plants that are well-functioning photosynthetic apparatus are 0.75 to 0.85 [46]. In this study, the obtained results showed that plants receiving calcium carbonate showed Fv/Fm values higher than 0.75, in contrast to those reported to control plants, to which the Fv/Fm values remained below 0.7. These results show us that the calcium carbonate is efficient in improving the protection against high light, protecting the photosynthetic apparatus. Opposed to it, Fv/Fm values below 0.7 are indicative of inactivation of RC of PSII, which associated with reduced activity of oxygen-evolving complex (OEC), reduces the photosynthetic potential of plants [45]. Furthermore, when CC was not applied, the PI_ABS_ values were also reduced. According to Oukarroum, Schansker and Strasser [47], low PI_ABS_ values are indicative of the occurrence of photoinhibition. Under photoinhition, the capacity of energy conservation is reduced [48,49]. Furthermore, the increase of PI_ABS_ values resultant of application of calcium carbonate indicates that the product is promoting photoprotection in the plants, which is corroborated with the increased Fv/Fm values and reductions of DIo/RC. The photoprotection allows a good functioning and better use of energy by photochemical apparatus.

In general, high DI_o_/RC values are more related with higher ABS/RC than the ET_0_/RC, which may lead to a decrease in PI_ABS_. Besides, PI_ABS_, Fv/Fm, DI_0_/RC, F_0_ and F_M_ are widely used to identify photoinhibition in plants because they are very sensitive parameters to changes in photosynthetic apparatus [43,45].

## 4. Materials and Methods

The experiment was conducted at the Fazenda Experimental da Universidade Federal do Espírito Santo (UFES), located in the municipality of São Mateus, State of Espírito Santo, Brazil, latitude 18°40′25″ S, longitude 40°51′23″ W (Figure 6). The climate is considered hot and humid, type Aw, with a dry season in Autumn–Winter and a rainy season in Spring–Summer, according to the Köppen classification [50].

The experiment was conducted in the months January to May 2020, and the climatic parameters—the rainfall, temperature and radiation data that were obtained by the automatic weather station of the National Institute of Meteorology, located in São Mateus—ES—can be seen in Figure 7A,B.

The experiment was conducted in a randomized block design in a 2 × 2 + 1 factorial scheme with eight repetitions. The sources of variation were two nozzle models (empty cone jet and flat jet), two application rates (100 and 150 L ha^−1^) and a control treatment. Along with the efficiency variables evaluated, a sunscreen was used for plants at a dose of 2.0 L ha^−1^, being the same recommended by the manufacturer, except for the control treatment. The product used is a mixed mineral fertilizer (Sombryt^®^, Lithoplant, Linhares/ES, Brazil), having in its composition calcium carbonate (18.5%), zinc oxide (0.5%) and adjuvants not described by the manufacturer.

A CO_2_ constant pressure type knapsack sprayer with a single lance was used for spraying. The working pressure for the empty cone jet and flat spray nozzles were 380 kPa and 413 kPa, respectively. The application rates were calibrated as a function of the working speed, controlled by a passometer; these ranged from 3.0 to 4.0 km h^−1^. The application height in relation to the seedlings was 50 cm.

The choice of spray nozzles for the tests was as recommended by the manufacturer (Table 5). The flat spray nozzle is made of ceramic core, resistant to all types of products, stream angle equal to 80° and produces medium droplets at low pressures and fine droplets at high pressures. The empty cone jet nozzle also made in a ceramic core, jet angle equal to 60°, recommended for both traction bar sprayers and hydro pneumatic sprayers, produces thin and extremely thin drops according to the work pressure.

The conilon coffee seedlings used were the genotype “153” from the Instituto Capixaba de Pesquisas, Assistência Técnica e Extensão Rural (Incaper), which is sensitive to heat/light stress. The seedlings were purchased from a certified commercial nursery, thus ensuring standardized seedlings according to nutritional and phytosanitary treatments commonly used in the nursery.

Coffee seedlings normally go through the process of acclimatization before being commercialized; those used in the experiment did not go through this process, except for the seedlings used in the control treatment. Therefore, seven days before the acclimatization process to which the seedlings would be subjected, the first application of calcium carbonate was made on the treatment seedlings. After the application of calcium carbonate, the seedlings remained in the nursery for seven days, after which they were removed from the nursery and taken to the Experimental Farm at UFES where the second part of the experiment was conducted. For the control treatment, the seedlings were acclimated 120 days after striking, remaining in the nursery for another 30 days, and then taken to the field.

For all treatments, the seedlings were planted in plastic pots with a volume of 5 L in which the substrate was nutritionally corrected, according to the planting recommendation; NPK formulation 00 15 00 was used for the correction.

Forty-two days after planting the seedlings in the pots, the second foliar application of sunscreen was performed in the treatments, except for the control.

To carry out the application processes, the methodology described in ISO Standard 22866 was followed [50]. This standard states that during the application process the temperature should be between 5 and 35 °C, for the wind speed, the standard allows a maximum of 10% of the measurements to be below 1.0 m s^−1^ and the wind direction within a limit of 90° ± 30° in relation to the spraying line. Thus, the ideal wind direction for applications to be made should be east–southeast (112.5°) and could be between east and southeast (90° and 135°). The climatic conditions of the site at the time of spraying were monitored by the automatic weather station of the Federal University of Espírito Santo, São Mateus Campus (Table 6).

A food colorant was added to the spray mixture containing the sunscreen (brilliant blue, dose 800 g ha^−1^ and a non-siliconized adjuvant based on balanced polymers specific for aerial applications with low volume of solution (0.3% *vv*^−1^)). In order to attest to the application efficiency in each of the treatments, water sensitive paper labels with dimensions of 76 × 26 mm were used to determine the coverage and density of sprayed droplets, and polyvinyl chloride (PVC) artificial targets with the same dimensions as the sensitive paper labels were used in the estimation of droplet deposits. The labels and the artificial targets were fixed on the third or fourth leaves of the coffee seedlings in each treatment

Quantification and characterization of impacts on each water-sensitive paper label were performed immediately after applying each treatment and drying the labels using a DropScope wireless system, consisting of application programs and a wireless digital microscope with a digital image sensor over 2500 dpi. This enables the estimated calculation of partially overlapping droplets from approximately 35 µm (Figure 8). The following parameters were evaluated: droplet density (droplets cm^−2^), coverage (%) and volume median diameter (µm). Water-sensitive paper labels have limitations regarding the detection of very fine droplets; however, this method was used due to its convenience in obtaining droplet spectrum results.

The artificial PVC targets were removed 30 min after the application of each treatment, in order to allow the evaporation of the solution water and keeping only the colorant, packed in plastic bags, properly identified and stored in a closed box to avoid exposure to solar radiation and potential degradation by oxidation of the pigment marker. The artificial targets were then transported to the laboratory, and the colorant removal was carried out by rinsing them with 50 mL of distilled water per sample. This washing step was performed with the artificial targets inside their own plastic bags.

Then the absorbance readings of these solutions were calculated in a *Thermo Scientific^®^* (Darmstadt, Germany) spectrophotometer, model Genesys 10 UV, set to measure the absorbance in the wavelength of 630 nm. The absorbance values were obtained by individual reading of each sample in the spectrophotometer and were converted into concentration (mg L^−1^) adopting the standard curve equation established by dilutions of 1/100, 1/200, 1/500, 1/1000, 1/2000, 1/5000 and 1/10,000 of the mixture sample collected in the mixing tank before application. The mass balance generated by the tracer dye deposits on the samples in relation to the initial concentration was used to estimate the deposition on the artificial targets. From the spectrophotometer reading, the calibration curve data and the area of the artificial targets, the amount of spray deposition per unit area was calculated in μg cm^−2^ (Equation (1)).
(1)$$\beta_{\mathrm{dep}}=\frac{\left(\rho_{\mathrm{sample}}{-\rho }_{\mathrm{white}}\right){\times F}_{\mathrm{caliber}}{\times V}_{\mathrm{dep}}}{\rho_{\mathrm{spray}}{\times A}_{\mathrm{leaf}}}$$
where: *β_dep_* is the deposition on the artificial targets, µg cm^−2^_;_ *ρ_sample_* is the spectrophotometer reading of the sample; *ρ_white_* is the spectrophotometer reading of the “white” test; *F_calibra_* is p calibration factor, µg L^−1^; *V_dep_* is the volume of dilution liquid, L; *ρ_spray_* is the sprayed concentration, g L^−1^; *A_leaf_* is the area of the artificial target.

Deposition data were normalized for comparison of treatments due to varying atmospheric conditions at the time of application. The normalized deposition was estimated according to the equation with Equation (2).
(2)$$\beta_{N}=\frac{\beta_{dep}\times 10^{5}}{Q\times \rho_{pulveriza}}$$
where $\beta_{N}$ is the normalized deposition (µg cm^−2^); $\beta_{dep}$ is the deposition on artificial targets (µg cm^−2^); *Q* is the application rate (L ha^−1^); $\rho_{pulveriza}$ is the sprayed concentration (mg L^−1^).

To evaluate the fluorescence induction curve (Kautsky curve), chlorophyll *a* fluorescence was measured on the 7th and 42nd days after the first and second sunscreen application respectively, for a total of two measurements.

To evaluate the fluorescence induction curve, a Handy-PEA portable fluorometer (Hanstech, King’s Lynn, Norkfolk, UK) was used during the morning period, between 07:00 and 10:00 am. The leaf clips (Hansateck, UK) used for dark-fitting the samples were positioned on the 3rd or 4th fully expanded young leaf selected from the apex Figure 9. Eight clips were attached per treatment.

The samples were dark-conditioned for a period of 30 min, enough time for the complete oxidation of the photosynthetic electron transport system. After this period, the leaves were exposed to a flash of red light (650 nm) with an intensity of 3000 µmol m^−2^ s^−1^. The fast fluorescence kinetics (Fo to F_M_) was recorded from 10 s to 1 s. The fluorescence intensity at 20 s (considered as Fo), 100 s, 300 s, 2 ms (FJ), 30 ms (FI) and maximum fluorescence or Fm was collected and used to obtain the parameters from the JIP-test, according to the Theory of Energy Flow in Biomembranes [40], using the software Biolyzer (Laboratory of Bioenergetics, University of Geneva, Switzerland). Normalizations of the OJIP transient fluorescence data were done according to Yusuf et al. [30]. All chlorophyll *a* fluorescence parameter evaluated in this study are described in Table 7.

The data were subjected to the Shapiro Wilk normality test, then the variance analysis done by the F test, where a significant difference was found; the means of the evaluated characteristics were compared using the Tukey test at 5% probability.

## 5. Conclusions

In conclusion, the application of calcium carbonate improved the photochemical performance of plants by increasing the performance index of PSII based on absorption, the maximum quantum yield for PSII primary photochemistry, the QA-reducing RCs per CS and decreasing the dissipated energy flux compared to those plants pre-acclimated and without the application of sunscreen, independent of the application rate and nozzle model used. Thus, it is evident that the product based on calcium carbonate and zinc oxide plays a relevant role in physiological maintenance, inducing photoprotection and reducing damage caused by abiotic stresses. These results give us another alternative to deal with abiotic stress in the field and consequently improve the initial development of seedlings. And finally, considering the risk of runoff of sunscreen applied at application rates of 150 L ha^−1^ using the flat spray nozzle, we suggest the use of a lower rate and empty cone nozzle in the application to conilon coffee seedlings.

## Figures and Tables

**Figure 1 plants-12-01467-f001:**
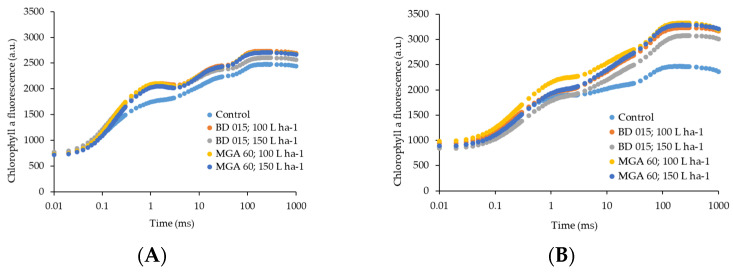
The OJIP chlorophyll *a* fluorescence transient (log time scale) in coffee leaves exposed to light stress. The measurements were made after 7 (**A**) and 30 days (**B**) of treatment. Before the measurements, leaves were dark-adapted for 30 min. Values are means for eight plants.

**Figure 2 plants-12-01467-f002:**
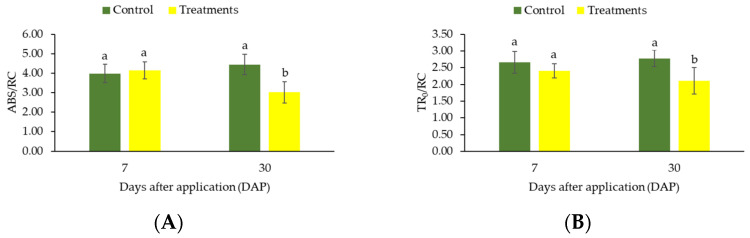
Changes in the ratios ABS/RC (**A**) and TR_0_/RC (**B**) in coffee leaves (genotype “153”) submitted to high light after application of CC. The measurements were made after 7 and 30 days of treatment. Before the measurements, leaves were dark-adapted for 30 min. Values are averages for eight plants. The data shows mean value ± standard errors. Means followed by different lowercase letters in each sampling (7 and 30 days) differ by Tukey’s test, at 5% probability.

**Figure 3 plants-12-01467-f003:**
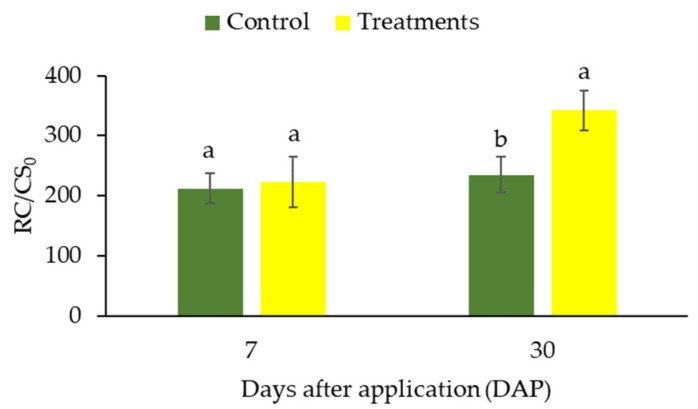
Changes in the ratios RC/CS0 (number of active reaction centers per cross section) in coffee leaves (genotype “153”) submitted to high light after the CC application. The measurements were made after 7 and 30 days of treatment. Before the measurements, leaves were dark-adapted for 30 min. Values are averages for eight plants. The data shows mean value ± standard errors. Averages followed by different lowercase letters in each sampling (7 and 30 days) differ by Tukey’s test, at 5% probability.

**Figure 4 plants-12-01467-f004:**
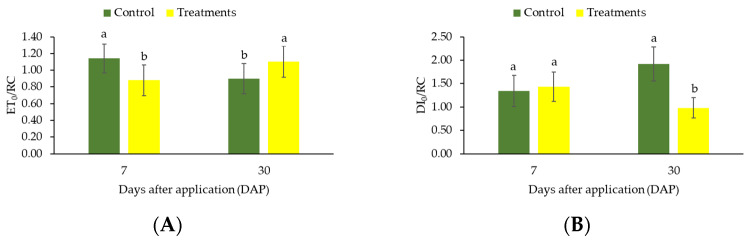
Changes in the ratios ET_0_/RC (**A**) and DI_0_/RC (**B**) in coffee leaves (genotype “153”) submitted to high light after application of CC. The measurements were made after 7 and 30 days of treatment. Before the measurements, leaves were dark-adapted for 30 min. Values are averages for eight plants. The data shows mean value ± standard errors. Means followed by different lowercase letters in each sampling (7 and 30 days) differ by Tukey’s test, at 5% probability.

**Figure 5 plants-12-01467-f005:**
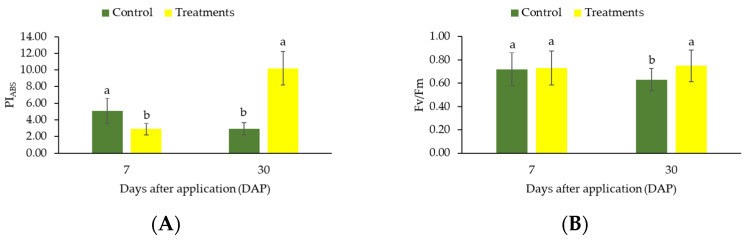
Changes in PIABS (**A**) values and Fv/Fm ratio (**B**) in coffee leaves (genotype “153”) submitted to high light after application of CC. The measurements were made after 7 and 30 days of treatment. Before the measurements, leaves were dark-adapted for 30 min. Values are averages for eight plants. The data shows mean value ± standard errors. Averages followed by different lowercase letters in each sampling (7 and 30 days) differ by Tukey’s test, at 5% probability.

**Figure 6 plants-12-01467-f006:**
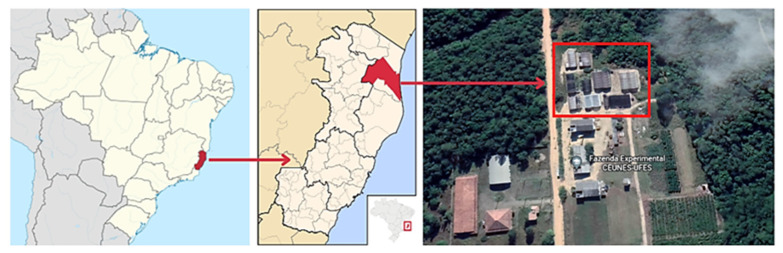
The test location and a brief overview of the acclimatization greenhouse marked with red rectangles.

**Figure 7 plants-12-01467-f007:**
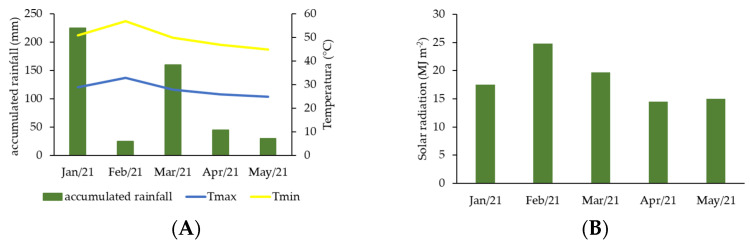
(**A**) Rainfall data (mm) and maximum and minimum temperature (°C) during the period of the experiment execution; (**B**) average monthly radiation data (MJ m^−2^) during the experiment execution period in the city of São Mateus—ES.

**Figure 8 plants-12-01467-f008:**
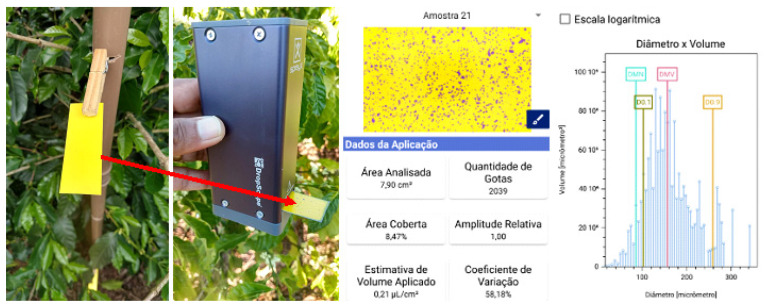
Attachment of water-sensitive labels to targets, DropScop^®^ Wirelles system for reading labels, and reporting of droplet deposition variables.

**Figure 9 plants-12-01467-f009:**
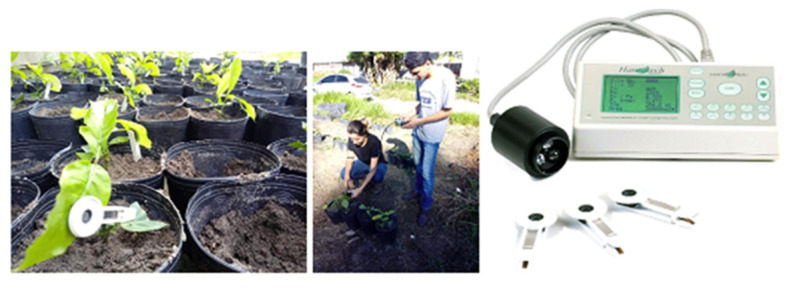
Schematic diagram of clip positioning and reading positioning with fluorometer, reading moment and detail Handy-PEA^®^ equipment.

**Table 1 plants-12-01467-t001:** Average values of deposited droplet density (droplets cm^−2^) after spraying with sunscreen on the leaves of conilon coffee seedlings at 7 (1st) and 30 (2nd) days after planting in pots in the field.

**Spraying**	**Nozzle**	**Application Rate (L ha^−1^)**	
		**100**	**150**
1 ª	MGA 60 01	32.29 ^bB^	38.42 ^bA^
	BD 015	66.93 ^aB^	80.99 ^AA^
CV = 31.39%.	W = 0.761 ^ns^	F_L_ = 1.249 ^ns^	
F_rate_ = 1.333 ^ns^	F_nozzle_ = 5.177 *	F_iteration_ = 2.488 *	
	**Nozzle**	**Application Rate (L ha^−1^)**	
		**100**	**150**
2 ª	MGA 60 01	25.75 ^bB^	30.87 ^bA^
	BD 015	61.24 ^aB^	78.85 ^aA^
CV = 35.02%.	W = 0.881 ^ns^	F_L_ = 1.425 ^ns^	
F_rate_ = 1.394 ^ns^	F_nozzle_ = 6.257 *	F_iteration =_ 2.661 *	

Averages followed by the same lower case letter in the column and capital case in the row do not differ by the Tukey test at 5% probability. CV = coefficient of variation; ns = not significant; * significant (*p*-value < 0.05); W = Shapiro Wilk test; F = Anova test.

**Table 2 plants-12-01467-t002:** Average values of deposited droplet coverage (%) after spraying the leaves of conilon coffee seedlings with sunscreen at 7th (1st) and 30th (2nd) days after planting in pots in the field.

**Spraying**	**Nozzle**	**Application Rate (L ha^−1^)**	
		**100**	**150**
1 ª	MGA 60 01	42.41 ^Bb^	49.85 ^bA^
	BD 015	66.52 ^aB^	71.12 ^aA^
CV = 38.88%.	W = 0.881 ^ns^	F_L_ = 1.559 ^ns^	
F_rate_ = 1.513 ^ns^	F_nozzle_ = 6.007 *	F_iteration_ = 2.772 *	
	**Nozzle**	**Application Rate (L ha^−1^)**	
		**100**	**150**
2 ª	MGA 60 01	39.94 ^Bb^	45.79 ^bA^
	BD 015	60.84 ^aB^	69.90 ^aA^
CV = 37.42%.	W = 0.902 ^ns^	F_L_ = 1.601^ns^	
F_rate_ = 1.224 ^ns^	F_nozzle_ = 5.987 *	F_iteration_ = 2.281 *	

Averages followed by the same lower case letter in the column and capital case in the row do not differ by the Tukey test at 5% probability. CV = coefficient of variation; ns = not significant; * significant (*p*-value < 0.05); W = Shapiro Wilk test; F = Anova test.

**Table 3 plants-12-01467-t003:** Average values of the volumetric median diameter of drops deposited (µm) on the leaves of coffee seedlings after spraying with the sunscreen on the leaves of conilon coffee seedlings at 7th (1st) and 30th (2nd) days after planting in pots in the field.

**Spraying**	**Nozzle**	**Application Rate (L ha^−1^)**	
		**100**	**150**
1st application	MGA 60 01	278.11 ^bA^	280.96 ^bA^
	BD 015	329.98 ^AA^	327.19 ^aA^
CV = 29.87%.	W = 0.711 ^ns^	F_L_ = 1.5759 ^ns^	
F_rate_ = 1.513 ^ns^	F_nozzle_ = 6.007 *	F_iteration_ = 2.772 ^ns^	
	**Nozzle**	**Application Rate (L ha^−1^)**	
		**100**	**150**
2nd application	MGA 60 01	284.2 ^bA^	285.4 ^bA^
	BD 015	322.1 ^ea^	320.2 ^aA^
CV = 30.09%.	W = 0.862 ^ns^	F_L_ = 1.771 ^ns^	
F_rate_ = 1.555 ^ns^	F_nozzle_ = 7.953 *	F_iteration_ = 1.442 ^ns^	

Averages followed by the same lower case letter in the column and capital case in the row do not differ by the Tukey test at 5% probability. CV = coefficient of variation; ns = not significant; W = Shapiro Wilk test; F = Anova test. * significant (*p*-value < 0.05).

**Table 4 plants-12-01467-t004:** Average values of droplet deposition (µg cm^−2^) on the leaves of coffee seedlings after spraying with the sunscreen on the leaves of conilon coffee seedlings on the 7th (1st) and 30th (2nd) days after planting in pots in the field.

**Spraying**	**Nozzle**	**Application Rate (L ha^−1^)**	
		**100**	**150**
1st application	MGA 60 01	3.45 ^bA^	3.57 ^bA^
	BD 015	4.79 ^aA^	5.01 ^aA^
CV = 19.98%.	W = 0.805 ^ns^	F_L_ = 2.229 ^ns^	
F_rate_ = 1.717 ^ns^	F_nozzle_ = 8.215 *	F_iteration_ = 2.772 *	
	**Nozzle**	**Application Rate (L ha^−1^)**	
		**100**	**150**
2nd application	MGA 60 01	3.88 ^bA^	3.95 ^bA^
	BD 015	4.92 ^aA^	5.11 ^aA^
CV = 30.09%.	W = 0.862 ^ns^	F_L_ = 1.771 ^ns^	
F_rate_ = 1.555 ^ns^	F_nozzle_ = 7.953 *	F_iteration_ = 1.442 ^ns^	

Averages followed by the same lower case letter in the column and capital case in the row do not differ by the Tukey test at 5% probability. CV = coefficient of variation; ns = not significant; * significant (*p*-value < 0.05); W = Shapiro Wilk test; F = Anova test.

**Table 5 plants-12-01467-t005:** Specification of the spray nozzles used.

Nozzle	Description	Pressure min/max (kPa)	Flow min/max. (L min^−1^)	Droplet Size
MGA 60	hollow cone	270/1040	0.39/0.68	Very fine
BD 015	flat fan nozzles	100/450	0.36/0.70	Fine

**Table 6 plants-12-01467-t006:** Values of minimum and maximum temperatures (TEM MIN/MAX), minimum and maximum relative humidity (RH min/max), minimum and maximum wind speed (VV MIN/MAX) in the Northeast direction after 7 and 30 days of planting the conilon coffee seedlings in pots in the field. Federal University of Espírito Santo, São Mateus Campus, Brazil.

Spraying	Temp min/max (°C)	RH min/max (%)	WS min/max (m s^−1^)
7 DAP	23.8/24.1	65/66	1.1/1.6
30 DAP	25.2/25.3	58/60	1.3/1.8

**Table 7 plants-12-01467-t007:** Abbreviations of the parameters, formulas and description of the data derived from the transient of Chla fluorescence.

Chl*a*F Parameters and Formulas	Description
Fo ≅ F_20µs_	Minimal fluorescence, when all PSII RCs are open
F_300µs_	Fluorescence intensity at 300 µs ms of OJIP
F_J_ ≅ F_2ms_	Fluorescence intensity at the J-step (2 ms) of OJIP
F_I_ ≅ F_30ms_	Fluorescence intensity at the I-step (30 ms) of OJIP
F_P_ ≅ F_300ms_ = F_M_	Maximal fluorescence at the peak P, when all PSII RCs are closed
V_J_ = (F_2ms_ − F_0_)/(F_M_ − F_0_)	Relative variable fluorescence at the J-step
V_I_ = (F_30ms_ − F_0_)/(F_M_ − F_0_)	Relative variable fluorescence at the I-step
ABS/RC = Mo·(1/Vj)·(1/φPo)	Absorption flux per active reaction center (RC)
TR_0_/RC = Mo·(1/Vj)	Trapped energy flux per RC (at t = 0)
ET0/RC = Mo·(1/Vj)·ψEo	electron transport flux per RC (at t ¼ 0)
DIo/RC = [(ABS/RC) − (TRo/RC)]	Dissipated energy flux per RC at t = 0
RC/CSo = φPo·(Vj/Mo)·(ABS/CS)	Q_A_-reducing RCs per CS
PI_ABS_ = [(RC/ABS) × (φP_0_)/(1 − φP_0_) × (ψE0)/(1 − ψE0)]	Performance index of PSII based on absorption
F_V_/F_M_ = TR_0_/ABS = φPo	maximum quantum yield for PSII primary photochemistry

## Data Availability

Not applicable.

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
