# Peer review of "Photosystem II Performance of Coffea canephora Seedlings after Sunscreen Application"

_plants, 2023, doi:10.3390/plants12071467_

Round 1

Reviewer 1 Report

In the manuscript titled "Spray Quality of Conilon Coffee Seedlings Submitted to Sunscreen Application", the authors evaluated the effect of two types of nozzles on the spray quality and protective effect of calcium carbonated based sunscreen against luminous stress. Below are my general remarks and a few suggested edits.

General comments:

·         Some details should be provided on the chlorophyll a fluorescence as a biomarker for photoinhibition in the introduction section.

·         Consistently use English to format tables and equations, otherwise use standard mathematical symbols.

·         In multiple comparison tests, use superscript letters to indicate statistical significancy.

·         There is a contradiction in the number of replicates used in the results and methodology sections. In the results section, it is indicated that the results represent the average observation from eight plants. However, in the material and methods section it is stated that six replications were included in the factorial design.

Suggested edits:

Line 25: However, we recommend using …

Line 41: … with desirable features in …

Line 59: However, excess light can damage the photosynthetic apparatus, …

Line 84: I suggest you consistently use the word ‘sunscreen’ instead of ‘grout’ which might be misinterpreted.

Line 88: … the photochemical effect of sunscreen …

Line 99 … empty cone spray nozzle, …

Line 100-102: The sentence is quite ambiguous. Consider rewriting it to provide a clear message.

Figure 1 should be annotated to indicate the OJIP phases.

Line 113-114: “The results were expected, meaning, the increase in droplet deposition with the increase in application rate.” Should be deleted.

Line 115: … values of sunscreen retained.

Line 136-137: …, variation in wind intensity might have contributed to the lower coverage observed for the MGA 60 01 nozzle.

Line 212: … light after application of CC.

Line 340: … because they are very …

Line 439: … 630 nm.

Line 446: ... were calculated.

Line 447-459: use English in the equations. Some parameters defined are not included in the equation. Please check if your equations are complete.

Table 7: Check the unites for Fk in the description column.

Author Response

Responser to Reviewer comments in attachment.

Reviewer 2 Report

Please find the attachment with a few corrections.

The manuscript Spray Quality of Conilon Coffee Seedlings Submitted to Sunscreen Application is descrbes influence of application of one sunscreen on photosyntesis  in conilon coffee.

The manuscript is very simple, however it can be classified rather as applied study. On the other hand when coffe seedlings are subjected to regular acclimation and it works without addition of chemicals, why to use sunscreen compounds? In this manuscript there is no explanation why the authors decided for such research. Anyway this manuscript  would probably fit better to other MDPI journals like Agronomy or Agriculture.

Author Response

Response to Reviewer comments im attachment.

Reviewer 3 Report

This seems to be a useful, well executed study.  My comments mostly refer to improving the clarity of the presentation.

Title:  rather than "spray quality" use "Photosystem II fluorescence of", and rather than "submitted to" use "after"

Line 14: omit "incidence of" and "radiation"

Line 20" rather than "interference", use "effects"

Line 84: define "grout"

Tables 1-4 define FL, Ftaxa, Fponta, FL

Fig. 1:  A vs. B is days 7 vs. 30?

Legends for Fig. 2-5: define ABS/RC etc in the legends as well as the text - the legends should be self-explanatory without reference to the text

Conclusions: spell out P1ABS etc. rather than using abbreviations

Author Response

Response to Reviewer comments in attachment.

Round 2

Reviewer 2 Report

The corrected version of the manuscript improved it essentially.

However, it will be good to write several introduction sentences at the beginning of Results, because now it is confusing to start reading Results with "The iteration between the nozzles and the application rates were significant in both moments of sunscreen application, indicating the dependence between the two factors..." A Reader does not know what is it about. Description of Figures 6, 8 and 9 should be improved. Individual photos in those figures should be described.

Author Response

Querido,

As sugestões e correções foram aceitas. Os comentários sobre as mudanças estão anexados.

Sinceramente
